# Drug Resistance and Endoplasmic Reticulum Stress in Hepatocellular Carcinoma

**DOI:** 10.3390/cells11040632

**Published:** 2022-02-11

**Authors:** Jaafar Khaled, Maria Kopsida, Hans Lennernäs, Femke Heindryckx

**Affiliations:** 1Medical Cell Biology, Uppsala University, 751 05 Uppsala, Sweden; jaafar.khaled@mcb.uu.se (J.K.); maria.kopsida@mcb.uu.se (M.K.); 2Pharmaceutical Biosciences, Uppsala University, 751 05 Uppsala, Sweden; hans.lennernas@farmbio.uu.se

**Keywords:** unfolded protein response, endoplasmic reticulum stress, liver cancer, drug resistance, transarterial chemoembolization, anthracyclins, tumor microenvironment

## Abstract

Hepatocellular carcinoma (HCC) is one of the most common and deadly cancers worldwide. It is usually diagnosed in an advanced stage and is characterized by a high intrinsic drug resistance, leading to limited chemotherapeutic efficacy and relapse after treatment. There is therefore a vast need for understanding underlying mechanisms that contribute to drug resistance and for developing therapeutic strategies that would overcome this. The rapid proliferation of tumor cells, in combination with a highly inflammatory microenvironment, causes a chronic increase of protein synthesis in different hepatic cell populations. This leads to an intensified demand of protein folding, which inevitably causes an accumulation of misfolded or unfolded proteins in the lumen of the endoplasmic reticulum (ER). This process is called ER stress and triggers the unfolded protein response (UPR) in order to restore protein synthesis or—in the case of severe or prolonged ER stress—to induce cell death. Interestingly, the three different arms of the ER stress signaling pathways have been shown to drive chemoresistance in several tumors and could therefore form a promising therapeutic target. This review provides an overview of how ER stress and activation of the UPR contributes to drug resistance in HCC.

## 1. Introduction

Hepatocellular carcinoma (HCC) is a primary liver tumor that contributes to over 550,000 annual deaths worldwide [1,2]. It usually develops in a background of chronic liver disease, which can be caused by alcohol-related liver diseases, chronic hepatitis infections, non-alcoholic fatty liver disease, and genetic mutations [3,4,5]. Each of these risk factors is characterized by a chronic perpetuation of liver injury that creates an inflammatory and a pro-tumoral microenvironment that sustains cancer cell proliferation and survival [6,7]. There is accumulating evidence that endoplasmic reticulum (ER) stress plays a pivotal role in chronic inflammation and carcinogenesis, and that in some cases, ER stress may even contribute to the initiation of these conditions [8,9,10]. Components of the ER are essential for folding proteins into their native three-dimensional (3D) conformations needed to perform their intended reactions and functions [8,9,11]. However, as protein translation increases in times of high cellular activity or stress, efficacy of ER components may decrease, leading to an accumulation of misfolded and unfolded proteins in the ER lumen. These misfolded proteins are often unable to leave the ER and may eventually trigger a signaling cascade that initiates the unfolded protein response (UPR) through three different ER stress-signaling branches: protein kinase RNA-like ER kinase (PERK), inositol-requiring enzyme 1 (IRE1α), and activating transcription factor 6 (ATF6). The UPR may also be elicited through other forms of stress that occur in the inflammatory microenvironment, as well as within rapidly proliferating tumor cells. Such triggers include an increased generation of reactive oxygen species (ROS), hypoxia, nutrient starvation, exposure to chemotherapeutic drugs and improper calcium homeostasis [12]. The activation of the UPR will primarily attempt to reduce the load of the ER, for instance by decelerating protein translation or by increasing the ER’s folding capacity. However, in cases of prolonged or severe ER stress, pro-survival and pro-apoptotic pathways will be initiated.

Hepatocellular carcinoma has several complex factors that contribute to its development and progression, in addition to mechanisms that can directly or indirectly interfere with treatment efficacy. One of the most alarming complications in HCC is occurrence of multidrug resistance. This means that efficacy of chemotherapeutic drugs become severely reduced due to cancer cells utilizing different biological tools to remove, convert, and/or disarm drugs intended to interfere with them [13,14,15,16]. This review focuses on how HCC cells harness ER stress to survive and adopt a drug resistant phenotype through several complex and intertwined resistance mechanisms (Figure 1).

## 2. Endoplasmic Reticulum Stress in Health and Disease

Under normal conditions, the ER is an organelle that manages synthesis and folding of proteins [8,9] while also functioning as a storage unit for calcium ions, lipid synthesis, and fatty acid oxidation [17]. When the cell’s need for protein folding exceeds the ER’s capacity to accurately fold proteins, an accumulation of unfolded and misfolded proteins will occur, thereby triggering the UPR. Activation of the mammalian UPR occurs through three ER stress pathways that ultimately regulate transcription of genes that influence protein folding, folding capacity, autophagy, and apoptosis [8]. Accumulation of unfolded or misfolded proteins is sensed via three transmembrane receptors: PERK, IRE1α, and ATF6, which remain inactive when they are bound to ER chaperone GRP78/BiP (78-kDa glucose-regulated protein) [8]. Activation of the different ER stress branches is thought to be based on two models, namely, the direct and indirect recognition models. The direct recognition model proposes that UPR activation arises when GRP78/BiP dissociates from the luminal binding domains of the PERK/ATF6/IRE1α receptors [8,11]. GRP78/BiP has a greater affinity for unfolded proteins than their ER stress receptor domain and therefore dissociates when unfolded protein concentrations increase [17]. During low levels of ER stress, activation of these pathways is correlated to increased translation of genes regulating ER chaperones, amino acid metabolism, redox reactions, autophagy, protein folding, and maturation, which is generally thought of as a pro-survival mechanism [8,9,10,11]. In comparison, high levels or prolonged ER stress have been shown to trigger apoptotic pathways via caspase-3, BCL-2 family member, apoptosome complex activation, and ferroptosis [17,18,19]. How and when different ER stress pathways exert their cytoprotective or their pro-apoptotic functions remains largely unknown. The duration and severity of ER stress seems to be a major contributor to the switch towards apoptosis, possibly by inducing changes in the conformational structure of IRE1α [17,20]. A second theory suggests that transcription factor E2F1 plays a role of a kill switch during late-stage ER stress, where expression levels of E2F1 will determine whether the cell locks onto survival or apoptotic pathways [17]. The threshold at which cells experience a severe and prolonged ER stress that would induce apoptosis varies among different cell lines, depending on their translational capacity (e.g., number of chaperones, ER size, and the amount of machinery for protein degradation) and differences in intrinsic sources of ER stress [21,22].

The rapid proliferation of tumor cells is accompanied by an acute increase of protein synthesis, which inevitably leads to activation of the UPR. It is therefore not surprising that actors of the UPR are increased in nearly all cancer types and that ER stress pathways seem to be affecting every hallmark of cancer [23,24]. Studies have shown that HCC cells hijack ER stress pathways to keep cells in a pro-survival signaling loop, while also interfering with components of the pro-apoptotic pathways [8,17]. Hypoxia and nutrient deprivation can directly activate the UPR [25] and have also been implicated as strong inducers of chemoresistance [13]. It has been further suggested that tumor cells modulate the UPR to aid in secreting pro-survival cytokines, growth factors, and other components, which consequently modulate cells in their immediate microenvironment to proliferate safely while suppressing an immune response [26,27]. To sustain their own metabolic demands and to adapt to a challenging and rapidly changing environment, cancer cells also reprogram their secretome to further support tumor function and induce chemoresistance. A thorough understanding of how ER stress pathways are intertwined with different mechanisms of drug resistance will therefore enable development of new chemotherapeutic candidates and optimize existing treatments in order to achieve better response.

## 3. Mechanisms of Drug Resistance

In recent decades, chemotherapy and targeted therapies have become principal modes of treatment against most types of cancer, including HCC. However, their efficacy is limited due to various inherent and acquired resistance mechanisms developed by cancer cells. This is specifically relevant in HCC, as most chemotherapeutics have limited efficiency due to the high intrinsic resistance of hepatic cancer cells [16]. Five different mechanisms have been described on how tumor cells acquire drug resistance: (1)—reduction of drug uptake, (2)—alteration of the drug target, (3)—induction of drug-detoxifying mechanisms, (4)—repair of drug-induced damage, and (5)—insensitivity to drug-induced cell death [14] (Figure 1). The three different arms of the ER stress signaling pathways are involved in the induction of chemoresistance throughout all these different mechanisms (Figure 2) [10].

### 3.1. Reduction of Drug Uptake and Enhanced Efflux

Reduction of drug uptake is a key mechanism known to cause cellular resistance of tumor cells against chemotherapeutic agents in different solid tumors [28]. Elevated expression of transport proteins expressed at the plasma membrane limits intracellular accumulation of anti-cancer drugs, either by pumping agents out of the tumor cells (enhanced efflux) or by blocking their uptake (reduced influx) [14]. The majority of these proteins belong to the mammalian adenosine triphosphate (ATP)-binding cassette (ABC) family of transporters. There are 48 types of functionally diverse human transporters, which have been divided into 7 distinct subfamilies (A-G) on the basis of their structural relatedness and domain organization [29]. Thus far, 16 types of ABC transporters are directly and/or indirectly linked to multidrug resistance in liver, kidney, pancreas, and other types of cancer (Table 1) [30,31,32]. The most extensively studied ABC transporters are the ABCB1 (permeability glycoprotein/MDR1), ABCC1 (multidrug resistance-associated protein-1, MRP1), and ABCG2 (breast cancer resistance protein (BCRP)) [32]. Their main ability is to recognize chemotherapeutic agents after their internalization within the plasma membrane and use the energy produced via ATP hydrolysis in order to expel drug molecules out of the cells, thereby decreasing bioavailability and increasing drug resistance [29,33]. Transportation of drugs across cells occurs through multiple processes, such as passive diffusion, facilitated or active transport, or pinocytosis. A major barrier that prevents drugs from accessing intracellular partitions in the plasma membrane is the solute carrier (SLC) family of proteins, which encode both passive and ion coupled carriers, as well as exchangers, which regulate 360 uptake carriers in the cell membrane [15]. Downregulation or inhibition of transporters may cause resistance to therapy due to reduced drug uptake or early/late impairment of endocytic pathways [14,34].

One of the most common drug resistance mechanisms in HCC is enhanced drug efflux through induction of ABC cassette transporters, such as MDR1, MRP1, and MRP2 [22,35,36]. Previous studies have shown that activation of ER stress pathways, specifically PERK and IRE1α, are important inducers of ABC transporter expression [23]. When it comes to PERK, Salaroglio et al. demonstrated through gene profiling analysis that high levels of PERK are found in human colon cancer cells resistant to chemotherapy [37]. This study further exposes that PERK forms an axis with nuclear receptor and transcription factor Nrf2, which was shown to directly regulate transcription of MRP1 [37]. In addition, this Nrf2/SHH signaling cascade is known to promote drug resistance in several HCC cell lines [35,38], as Nrf2 activity upregulates multidrug-resistant efflux pumps [35,39,40]. In addition, Nrf2 has also been identified as a transcription substrate of PERK, thus further strengthening the interaction between Nrf2 expression and ER stress signaling [41]. The ATF4/PERK pathway also interacts with the long non-coding RNA (lncRNA) ZFAS1 signaling pathway, which has been shown to be important in sorafenib resistance [42]. Sorafenib itself may contribute to ZFAS1 activation by activating PERK/ATF4-pathways in drug-resistant HCC cells and inhibiting the PERK-signaling pathway, which has been suggested to re-sensitize cells to sorafenib [42]. This suggests that specific PERK inhibitors could provide an attractive therapeutic target to enhance efficacy of sorafenib treatment in HCC [42]. The strong link between PERK and drug response has led to several studies using PERK inhibitors in preclinical studies; however, careful evaluation is warranted, as there have been critical issues with specificity of some of these inhibitors [43].

Although IRE1α’s relationship to ABC transporters has not been clearly established in HCC, Gao et al. (2020) found that the IRE1α–XBP1 axis participates in activating efflux pumps ABCC1 and ABCG2 in colon cancer cells resistant to 5-fluorouracil [44]. Treatment with 5-fluorouracil induces activation of IRE1α–XBP1 and increases expression of ABC transporters. However, further research is necessary to clarify the role of IRE1α and its relationship to enhanced drug efflux in HCC. One important aspect of IRE1α’s endoribonuclease activity is that this is not only responsible for cleaving XBP1, but several studies have shown that IRE1α is also able to cleave and regulate miRNAs [45,46,47]. ER stress has therefore been shown to contribute to the miRNA imbalance in inflammation [47] and cancer [48]. This could be particularly interesting in the context of drug resistance, as Shi et al. has previously shown that upregulation of miRNAs may confer resistance to 5-fluorouracil in HCC cell lines [49]. Through a CCK-8 assay, researchers found that ectopic expression of MiR-141 can directly induce chemoresistance in HepG2, SMMC-7721, and Huh7 cell lines [49]. Moreover, MiR-141 facilitates Nrf2 pathway activation by targeting Kelch-like ECH-associated protein (Keap1), which under healthy conditions mediates ubiquitination and degradation of Nrf2. Thus, Shi et al. (2015) further demonstrated that MiR-141 deregulates Keap1 protein stability, thereby inducing Nrf2, which may in turn result in overexpression of ABC cassette transporters [49]. On the other hand, miR-122 specifically targets the membrane transporter SLC7A1, which is associated with resistance to sorafenib. miR-122 upregulation can potentially knock down SLC7A1 expression and re-sensitize HCC cells to sorafenib treatment [50]. On the basis of previous results, ER stress seems to act as a regulator of miRNAs, which could potentially influence drug response through different mechanisms that involve either drug efflux (such as miR-131) or drug uptake (such as miR-122). However, further research is necessary to identify the exact pathways that could be relevant in HCC and underlying mechanisms that regulate these miRNAs, as well as their target genes.

The PERK/ATF4 branch of the ER stress pathways has also been implicated in drug uptake. Gao et al. uncovered that YAP/TAZ transcription factors play a crucial role in sorafenib resistance in HCC by inducing SLC7A11 expression, which is a core carrier that sustains intracellular glutathione homeostasis [51]. This allows HCC cells to overturn sorafenib-induced ferroptosis. Simultaneously, protein stability, nuclear localization, and transcriptional activity of ATF4 is upheld by YAP/TAZ activation. In turn, ATF4 also collaborates with YAP/TAZ to activate SLC7A11 expression, thus further enhancing drug transport over the cellular membrane. Hence, YAP/TAZ represses ferroptosis and consequently contributes to sorafenib resistance in HCC in an ATF4-dependent manner. This further underlines the potential value of rewiring approaches that rely on ATF4 and YAP/TAZ activation as a means to counter chemotherapeutic resistance in HCC [51].

### 3.2. Alteration of the Drug Targets

Many anti-cancer drugs must undergo metabolic activation to exert their cytotoxic effects. Cancer cells can therefore develop resistance by altering expression of specific enzymes involved in drug metabolism. One such example is DNA topoisomerase, which is a family of nuclear enzymes that regulate DNA topology and is recognized as the primary targets of chemotherapeutic drugs, including anthracyclines (e.g., doxorubicin) [52]. Activation of the UPR has been shown to reduce topoisomerase IIα protein levels [53] and thereby decrease sensitivity to topoisomerase-targeted drugs, such as doxorubicin [54]. In addition, exposure of cells to etoposide, a topoisomerase II inhibitor, triggers mild IRE1α phosphorylation, without triggering other pathways of ER stress [55], which further suggests an interaction between ER stress signaling pathways and potential alteration of drug targets. However, other studies have argued that activation of the PERK branch of the UPR is the major contributor to UPR-induced etoposide resistance, regardless of changes in topoisomerase IIα protein expression [56]. Therefore, more research is necessary to confirm whether the UPR-induced decrease in topoisomerase IIα is responsible for increased resistance to chemotherapeutics in HCC, or whether this is a result of another ER stress-dependent drug resistance mechanism.

### 3.3. Induction of Drug-Detoxifying Mechanisms

Drug inactivation can play a major role in the development of resistance to chemotherapeutic drugs. This can be achieved, for example, through conjugation of chemotherapeutic drugs to glutathione by glutathione S-transferases [56]. Studies have shown that autophagy can increase expression of glutathione transferases, thereby directly contributing to chemoresistance [57]. Autophagy is an intracellular lysosome-mediated degradation pathway used for recycling and eliminating proteins and protein aggregates, an essential protective mechanism that is activated during ER stress [58]. Both ER stress and autophagy systems are dynamically interconnected, with ER stress pathways both stimulating or inhibiting autophagy. Autophagy protects cancer cells from chemotherapy by repressing apoptotic signals, as well as by triggering activation of glutathione transferases. These enzymes are involved in catalyzing nucleophilic addition of glutathione to chemicals with an electrophilic functional group (including products of oxidative stress, as well as directly binding to chemotherapeutic agents) [59]. Autophagy in HCC can thereby contribute to tolerance of chemotherapeutic drugs, such as oxaliplatin, via regulation of ROS levels or by direct glutathione conjugation. One study specifically showed that knockdown of glutathione transferase mu-1 in MHCC97-H and Huh-7 cells increases resistance to oxaliplatin and sorafenib, as well as the fact that oxaliplatin-induced autophagy can be downregulated by silencing this enzyme [57]. Furthermore, Shi et al. demonstrated that sorafenib exposure in HCC cells upregulates IRE1α pathway signaling, thereby inducing autophagy [60]. This study also revealed that when autophagy was suppressed, HCC cells were re-sensitized to ER stress-induced cell death, both in in vitro as well as in in vivo studies [60]. These findings underline the role of sorafenib-related ER stress in triggering autophagy, which could be directly or indirectly related to the activation of glutathione transferases [60]. There have also been studies showing a potential direct link between ER stress pathways and glutathione transferases, which could contribute to drug-detoxifying mechanisms in drug resistant tumors. For instance, glutathione S-transferase P-mediated S-glutathionylation has been indicated to regulate activities of a number of redox-active ER proteins, including ER stress chaperone BiP [61]. The S-glutathionylation of BiP has then been proposed to contribute to bortezomib resistance in multiple myeloma cells [61]; however, more research is necessary to confirm this in HCC.

As mentioned previously, a recent study by Gao et al. showed that ATF4 induces activation of solute carrier SLC7A11, a cystine importer that plays a central role in the synthesis of glutathione [51]. Glutathione synthesis participates in suppression of ferroptosis and apoptosis in HCC cell lines in order to achieve cellular homeostasis. Moreover, nuclear import of ATF4 was found to be regulated and stabilized by YAP and TAZ transcription factors. In fact, YAP/TAZ and ATF4 proteins jointly trigger genes associated with antioxidant mechanisms, including SLC7A11 [51]. Antioxidant mechanisms may hamper efficacy of chemotherapy by scavenging ROS and free radicals, which are known to directly contribute to cytotoxic effects of many chemotherapeutics, including doxorubicin [62]. Recent studies have suggested a link between activation of the IRE1α pathway and generation of ROS, thus further suggesting that drug-detoxifying mechanisms could contribute to an ER stress-dependent effect on drug resistance [22].

### 3.4. Repair Mechanisms

The majority of chemotherapeutics contain DNA-damaging reagents, which means tumor cells are able to develop resistance against chemotherapy by repairing DNA damage [63]. DNA damage response (DDR) is a complex signal transduction pathway that is able to repair various endogenous and exogenous DNA lesions that accumulate in cells [64,65]. Phosphorylation, ubiquitination, and sumoylation are part of the critical post-translational alterations that occur on DNA damage response genes, the latter being necessary for proper repair [66]. Although DNA repair protein impairment usually leads to apoptosis, cells may also experience genomic degradation according to the degree of damage [65]. Several genes have been reported to regulate DNA damage and repair in HCC, such as mitogen-activated protein kinase–extracellular signal-regulated kinase (MAPK-ERK), BRCA, p53, c-MYC, and interleukin-6 (IL-6), thus leading to intensive recovery from chemotherapy-induced DNA damage and thereby conferring resistance to tumor cells [63,67,68].

Oshi et al. found that activation of BRCA1 and BRCA2 was correlated with a high DNA repair score in HCC, which was also related to poor survival, increased cell proliferation, high intra-tumoral heterogeneity, and mutational burden [63]. RuvB-like2, which is involved in DNA damage detection and repair, enhances cell proliferation and is overexpressed in HCC, leading to a poor prognosis [63]. This gene was also shown to be closely related to the ER stress degradation (ERAD) mechanism by negatively mediating ER stress response proteins [69].

P53 is a tumor suppressor gene that plays an essential role in regulating cell cycle arrest, DNA-repair, and apoptosis as a result of DNA damage. It facilitates DNA repair by pausing the cell cycle, thereby allowing time for repair machineries to restore stability [70]. In addition, p53 takes on diverse roles that directly impact activity of various DNA repair systems [71]. Activation of p53 is tightly regulated by a complex web of pathways that control its post-translational modification, sub-cellular localization, and degradation [64,71]. Although the UPR has been suggested as an essential player in regulating p53 [72], the exact mechanism is not fully understood, since ER stress can both activate [73] or destabilize p53 [74,75]. Activation of p53 during ER stress was mainly associated with increased apoptotic cell death acting through NF-κB signaling pathways [73]. However, other studies have shown that p53 adopts a mostly cytoplasmic distribution during ER stress, suggesting an inhibition of its function [74]. The UPR regulates glycogen synthase kinase 3β, which is essential for regulating p53 and cyclin D1 degradation in the event of early ER stress [74]. Further studies have suggested that ER stress induces destabilization of p53 and therefore prevents cells from p53-dependent apoptosis, which could form an important mechanism of resistance to chemotherapy [75]. This is mediated, at least in part, through increased cytoplasmic localization of p53 as a result of phosphorylation at serines 315 and 376 by increased activity of glycogen synthase-3 β. Although molecular mechanisms remain largely unknown, BiP’s mRNA is also known to interact with p53. This interaction results in inhibition of BiP protein synthesis and leads to a decreased interaction between BiP and Bcl-2-interacting killer protein, a pro-apoptotic member of the Bcl-2 family [76]. The downstream target of p53, namely, p21, has further been shown to be regulated by PERK [77] and CHOP [78]. In addition, p21 induction is thought to play an important role in the response to ER stress, as p21 is a pro-survival effector of ATF4 [79]. The above-mentioned results on the role of p53 during ER stress focus on the full-length p53 protein without addressing the expression of its isoforms. These isoforms exist to facilitate the protein’s response to different stimuli, as they can modulate activity of the full-length protein or affect different pathways [80]. One of these p53 isoforms is p47, whose expression is specifically induced by PERK during ER stress [81]. Overall, these different studies suggest a tight link between p53, ER stress, and DNA repair mechanisms that needs to be further elucidated to truly understand its role in mediating drug resistance in HCC.

The MAPK signaling pathway has also been reported to contribute to DNA repair in HCC [63]. MAPK signaling pathways are activated in response to extracellular signals, such as growth factors, cytokines, and ER stress [4]. The UPR promotes ASK1-MKK4/7-JNK expression and also induces ERK1/2 activity. Nonetheless, MEKK4-MKK3/4/6-p38 signaling also regulates UPR through CHOP and ATF6 p38-related phosphorylation. IRE1α has been shown to induce JNK and ERK1/2 and modulate the oligomerisation of TRAF2, which in turn can trigger the ASK1-MKK4/7-JNK-pathway [82], thereby suggesting an important role of ER stress in inducing MAPK-dependent DNA repair mechanism, which could then contribute to drug resistance.

### 3.5. Insensitivity to Drug-Induced Cell Death

Another important mechanism that contributes to drug resistance is inhibition of cell death. Expression of prognostic ER stress proteins such as GADD34, eIF2α, CHOP, and ATF4, being regulated by PERK, were shown to be involved in apoptosis, as well as in chemo-responsiveness [37,83]. Meanwhile regarding ferroptosis, another type of cell death [84], recent studies have shown there is a cross-talk between ER stress and the lipidome that mediates chemoresistance when responding to anthracyclines, such as doxorubicin and idarubicin [18]. The lipid membrane composition was established to be the main factor affecting drug fluidity and membrane permeability. In addition, lipid membrane structure also shapes activation of drug transporter-regulated efflux mechanisms related to multidrug resistance [85]. Therefore, the study suggests that in the interconnection between ER stress pathways and the lipidome, there is an overall shift in the balance among various lipid types, which is associated with ER stress [18]. For example, the imbalance of PUFA-containing lipids in comparison to saturated lipids and cholesterol contributes to ER stress, which can result in the repression of apoptosis and ferroptosis [18].

Furthermore, ER stress receptor of activated protein kinase (RACK1) was found to be critical in inducing IRE1α signaling response to sorafenib in HCC cells [86]. Although IRE1α activation was shown to favorize ER stress-induced apoptosis via JNK, RACK1 dysregulation was found to be responsible for IRE1α’s high phosphorylation, thereby promoting XBP1 splicing and preventing sorafenib-induced cell death [23]. Indeed, overexpressed RACK1 was shown to prevent apoptotic action of sorafenib in HCC cells by upregulating XBP1, which consequently decreased when RACK1 was depleted [86]. Both RACK1 and IRE1 were found to be intercellularly interacting and were co-located in the cytoplasm. Silencing of RACK1 strongly inhibited IRE1 phosphorylation following treatment with either tunicamycin or sorafenib, which resulted in UPR suppression. According to the findings of Zhou et al., there is a crosstalk between RACK1 and IRE1, suggesting that they cooperate to regulate IRE1 signaling activity in response to sorafenib-dependent UPR activation, which in turn controls ER-mediated cell death following treatment [86].

In addition, as previously mentioned, PERK is responsible for regulating Nrf2 [41], which besides activating ABC transporter efflux pumps [39], has also been reported to promote anti-apoptotic signals [87]. The role of Nrf2 in blocking cell death was confirmed as inhibition of Nrf2 restored sensitivity to apoptosis and reversed chemoresistance in Bel-7402/5-FU cell lines [88]. It was further established that miR-144 is responsible for Nrf2 mRNA degradation by directly targeting the Nrf2 3’untranslated region in chemo-sensitive HCC cells and the fact that resistant cell lines were characterized by a dramatic loss of miR-144 [88]. In an effort to further examine the influence of Nrf2 on cell death, another study suggests that its elicitation might be related to mitochondrial DNA depletion in HCC cell lines [89]. It has been previously demonstrated that there is an underlying relationship between mtDNA and ER stress, suggesting that any disturbance in the ER could have an impact on mtDNA and vice versa [90,91,92]. Investigating the crosstalk between ER stress and mtDNA depletion could then provide precious insights to further assess chemoresistance mechanisms in HCC in relation to cell death [92], especially as mtDNA depletion was also found to trigger other markers such as survivin, Bax, and Bcl2, which, when imbalanced, hinder pro-apoptotic signals [89].

Other studies suggest that IRE1α and PERK can induce STAT3 and NFκB [93,94,95], which are known to transcriptionally overexpress cell death-inhibiting proteins, such as Bcl2, caspase-8 inhibitor c-FLIP, MCL1, and IAP [96,97]. Another possibility is activation of transcription factor ATF6, which is also shown to confer chemoresistance in HCC cell lines by impeding ER stress-induced cell death and promoting cell survival [98]. ATF6 was observed to form an axis with protein disulfide isomerase P58 [99], thereby restricting cell death under oncogenic transformation, by suppressing the PERK-CHOP pro-apoptotic pathway [21]. At the same time, PERK has been shown to contribute to synthesis of eIF2α, which supports cancer cell survival. Therefore, CHOP has been identified as a promoter of hepatocarcinogenesis [100,101], despite its well-established pro-apoptotic function in physiological conditions [102].

Moreover, as mentioned earlier, an intrinsic link between ER stress and miRNAs has been suggested [48]. Addressing this deregulation in depth, research found that in addition to triggering ABC transporters, over-expression of miR-141 blocked cytotoxic and apoptotic activity of 5-fluorouracil in several HCC cell lines [49,88]. In parallel, when miR-141 was knocked down, cells regained susceptibility to apoptosis triggered by 5-fluorouracil. MiR-141 would therefore have the capacity to confer chemoresistance to cancer cells by disrupting their sensitivity to 5-fluorouracil-induced cell death [49]. This further displays the interconnection between ER stress, miRNA, and cell death that could contribute to chemoresistance in HCC.

## 4. Role of the Tumor Microenvironment

The tumor microenvironment is composed of cancer and stromal cells, surrounded by vast amounts of extracellular matrix proteins, and characterized by an abnormal and dysfunctional vasculature. This tumor microenvironment is not a passive bystander in the hepatocarcinogenic process but actively fuels and regulates tumor progression, metastasis, and drug response [6].

Tumor hypoperfusion, hyperpermeability, and leakiness of the abnormal tumoral vasculature, along with hypoxia, nutrient deprivation, acidic conditions, and high interstitial pressure, can contribute to reduced response to chemotherapeutics. In addition, these are all common triggers of the UPR (Figure 3) [6,103,104,105]. Hypoxia has been shown to induce resistance to sorafenib, cisplatin, 5-fluorouracil, gemcitabine, adriamycin, and 6-thioguanine in several HCC cell lines [106]. Furthermore, hypoxia and nutrient deprivation are strong independent UPR inducers by upregulating GRP78/BiP expression levels [107]. It has been shown that knocking down GRP78/BiP with short hairpin RNA can improve response to cisplatin treatment under severely hypoxic conditions [107]. These results indicate a clear link between ER stress signaling, hypoxia, and drug response, yet more research is necessary to confirm this in HCC.

Sustained liver damage and chronic inflammation leads to activation of hepatic stellate cells, which increases deposition of extracellular matrix proteins, such as collagen and fibrinogen. The abundance of extracellular matrix proteins increases liver stiffness, thereby altering bio-mechanical properties and inducing mechanical stress in different hepatic cell populations [108]. The increased stiffness of a cirrhotic liver also leads to hypoxia, acidosis, nutrient deprivation, altered bio-mechanical properties, and generation of ROS, which can all directly induce the UPR [109]. It is therefore not surprising that increased liver stiffness has been correlated with decreased drug response of HCC cells embedded in a 3D matrix consisting of collagen and/or fibrinogen [110,111]. However, whether this is a result of an increased activation of the UPR still needs to be confirmed.

Cancer-associated fibroblasts (CAFs) and activated stellate cells are key players in tumor–stroma interactions and are major mediators in carcinogenesis, tumor progression, and chemoresistance [112,113]. Studies have shown that ER stress plays a crucial role in activation of hepatic stellate cells [114,115] and that blocking the IRE1α–XBP1 pathway can significantly reduce liver fibrosis and tumor burden in several animal models for cirrhosis [47] and HCC [22]. Jia et al. sought to provide a deeper investigation regarding the involvement of CAFs in epithelial–mesenchymal transition (EMT) in HCC [116], a process that is known to contribute to drug resistance in HCC and many other solid tumors [117,118]. Proteomic analyses identified that transglutaminase 2 (TG2) is substantially overexpressed in HCC cells that have obtained an EMT phenotype. Increased TG2 activity supported EMT in HCC cells, while TG2 depletion significantly decreased CAF-induced EMT [116]. Moreover, TG2 activity was improved once HCC cells were triggered by IL-6 during EMT, and suppressing IL-6/STAT3 signaling lowered TG2 activity. Consequently, H-CAFs promote EMT in HCC cells mediated by IL-6, activating the IL-6/IL6R/STAT3 pathway and thereby inducing TG2 [116]. As ER stress is an important factor in activating CAFs and stellate cells, it could therefore be proposed that this would then contribute to an increased drug resistance through CAF-induced EMT.

Another major component of the tumor microenvironment are tumor-associated macrophages, which play a pivotal role HCC [119,120]. Macrophages are heterogeneous by nature, as they actively engage in both induction and resolution of inflammation. The ability of macrophages to be reprogrammed is an active area of research, especially in HCC, where they have been reported to play a dual role in promoting or inhibiting tumor progression, depending on their polarization state. Firstly, classically activated macrophages, which develop a proinflammatory Th1 immune response and exert an anti-tumoral activity. Secondly, alternatively activated macrophages, or tumor-associated macrophages (TAM), which display an anti-inflammatory Th2 immune response and exert a tumor-promoting activity. The balance between these anti-tumoral and pro-tumoral macrophages has a significant impact on determining the response to chemotherapy. For instance, studies have shown that TAMs modulate resistance to oxaliplatin by inducing autophagy in HCC cells [121]. In addition, actors of ER stress pathways have been shown to play an important role in regulating the polarization state of hepatic macrophages [122]. Studies have shown that activation of ER stress pathways is associated with an increased anti-tumoral and pro-inflammatory phenotype, mainly through activation of PERK and CHOP [122,123,124]. By mediating the macrophage’s polarization state and function, the UPR could therefore influence chemotherapeutic response in HCC. Furthermore, studies have shown that IRE1α mediates release of inflammatory extracellular vesicles by hepatocytes in mouse models for non-alcoholic steatohepatitis [125]. It remains to be elucidated as to whether this also affects the inflammatory response in HCC and contributes to alterations in drug response.

However, it has become increasingly clear that UPR signaling plays an important role in immunity and inflammation. The three different arms of the UPR pathways have been involved in activating different inflammatory cell types, including tumor-associated macrophages, monocytes, dendritic cells, and tumor-associated neutrophils [126]. The latter also play an important role in orchestrating drug resistance in HCC [127]. Interestingly, a recent study demonstrated that neutrophils can drive PERK-mediated apoptosis in cancer cells through secretion of arginase-1 [128]. The neutrophil’s secreted arginase will lead to arginine deprivation in cancer cells, thereby inducing ER stress and resulting in apoptosis of tumor cells. In addition, as different inflammatory cell populations become activated during liver damage, they are known to generate large amounts of growth factors, which in itself can contribute to therapeutic resistance. In resistant HCC cells, enhanced autocrine generation of growth factors, such as interleukin (IL)-1, IL-4, IL-6, and IL-8, was observed in comparison to tumor cells responsive to treatment [129,130]. The IRE1α–XBP1 pathway was found to mediate IL-6 expression and subsequently support tumor cell proliferation in HCC [131]. In human HCC tissues and cells, XBP1 splicing levels and IL-6 concentrations were elevated and favorably correlated to one another [131]. Furthermore, secretory levels and expression of IL-6 were lowered after pharmacological inhibition of IRE1α’s splicing activity, thereby hindering XBP1s direct binding to IL-6 and limiting expression of this interleukin. The enhancing effect of IRE1α–XBP1 signaling in Hep3B cell proliferation was alleviated through inhibition of IL-6–STAT3 signaling by tocilizumab. As a result, by mediating activation of the IL-6-STAT3 signaling pathway, the IRE1α–XBP1 axis was shown to play a critical role in HCC carcinogenesis [131].

## 5. Discussion

Hepatocellular carcinoma (HCC) is one of the most common and deadly cancers worldwide. It is usually diagnosed in an advanced stage and is characterized by a high intrinsic drug resistance leading to limited chemotherapeutic efficacy [13,16]. There is therefore a vast need for understanding underlying mechanisms that contribute to drug resistance and for developing therapeutic strategies that could overcome this process [132]. One important mechanism that contributes to drug resistance is activation of endoplasmic reticulum (ER) stress pathways, which then leads to induction of the unfolded protein response (UPR) [10,21,37]. The UPR is a conserved cell survival strategy and stress response, initiated when a cell’s need for protein synthesis exceeds the ER’s capacity to ensure accurate protein folding [12,21,92]. In such cases, accumulation of misfolded or unfolded proteins, known as ER stress, is sensed through three ER transmembrane proteins (IRE1α, PERK, ATF6), which activate the UPR with the goal of re-establishing normal ER function or inducing apoptosis. Actors of the UPR pathways are activated in the majority of cancers [133], and their expression has been correlated with poor prognosis in HCC [22,23], as well as decreased response to chemotherapeutics [10]. Tumor cells obtain resistance to chemotherapeutics mainly through five different mechanisms: reduction of drug uptake, alteration of the drug targets, induction of drug-detoxifying mechanisms, repair of drug-induced damages, and insensitivity to drug-induced cell death [16,28,65]. In addition, the tumor microenvironment has also been shown to play a pivotal role in determining response to chemotherapeutics. In this review, we highlighted how the three arms of the UPR are highly connected to the different mechanisms that contribute to chemoresistance in HCC.

## Figures and Tables

**Figure 1 cells-11-00632-f001:**
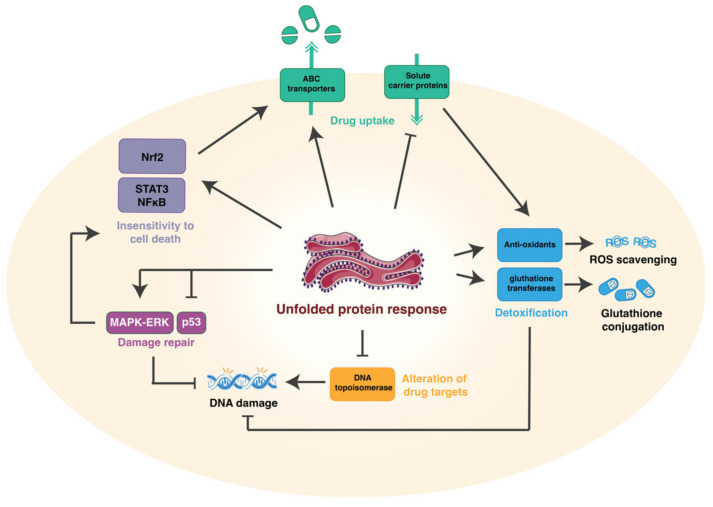
There are five mechanisms on how tumor cells acquire drug resistance. Firstly, reduction of drug uptake by increased efflux and decreased influx of chemotherapeutics through ABC transporters and solute carrier proteins, respectively. Secondly, alteration of the drug targets, for instance by decreasing protein expression of DNA topoisomerase, an important target of doxorubicin, which is responsible for inducing DNA strand breaks during doxorubicin treatment. Thirdly, by induction of drug-detoxifying mechanisms, such as scavenging of reactive oxygen species (ROS) or by nucleophilic conjugation of glutathione to the active site of chemotherapeutics, which is mediated by glutathione transferase enzymes. Fourthly, by repairing drug-induced damages, such as DNA damage, which can be directly induced by chemotherapeutics or through oxidative stress. Lastly, by inducing insensitivity to cell death through activation of several pathways, including NFkB, STAT3, and Nrf2. These five molecular mechanisms are heavily intertwined, thereby often accelerating the drug-resistant phenotype of hepatocellular carcinoma.

**Figure 2 cells-11-00632-f002:**
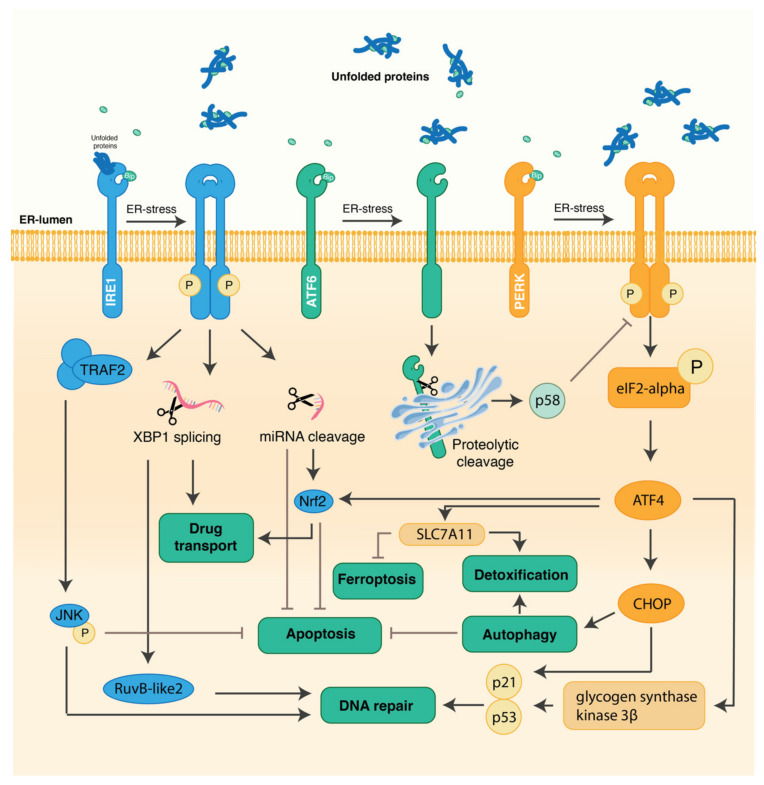
The accumulation of unfolded proteins in the endoplasmic reticulum lumen induces ER stress, leading to the dissociation of BiP from IRE1α, ATF6, and PERK. This activates the three pathways, resulting in the unfolded protein response, which in turns stimulates many underlying pathways and mechanisms that contribute to increased chemotherapeutic resistance in HCC. IRE1α mainly leads to DNA repair and inhibition of apoptosis trough the activation of the TRAF2/JNK pathway, but also the alteration of drug transport through XBP1 splicing and Nrf2 activation. ATF6 on its turn activates p58 via proteolytic cleavage. Finally, PERK is mainly responsible for the mechanisms activating autophagy through the eif2-alpha/ATF4/CHOP axis and DNA repair (eif2-alpha/ATF4 pathway). It also inhibits ferroptosis via the eif2-alpha/ATF4 pathway.

**Figure 3 cells-11-00632-f003:**
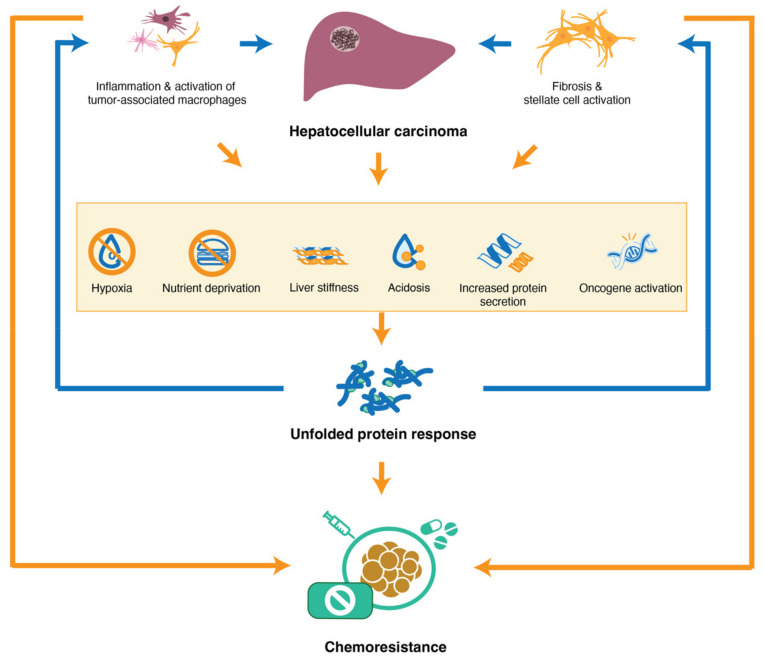
Tumor hypoperfusion, along with low oxygen levels, depleted nutrition, low pH (acidosis), increased liver stiffness, and an overall increased need for protein translation by rapidly proliferating tumor cells and recruitment of inflammatory cells, as well as activation of oncogenes, all induce the UPR inside tumor cells and in its microenvironment. This activation of the UPR will amplify the pro-tumoral inflammatory response and further increase activation of stellate cells, leading to fibrosis and deposition of ECM, thus inducing a vicious circle that further fuels ER stress pathways and contributes directly and indirectly to increased chemoresistance in HCC.

**Table 1 cells-11-00632-t001:** ATP-binding cassette ABC efflux pumps involved in enhanced drug efflux and reduced drug uptake stratified according to their TMD folds.

TMD Fold	Subfamily	Gene	Function	Upregulation in Cancer
Type IV	ABCB	ABCB1 (P-gp)	Drug efflux and regulator of lipids and steroids homeostasis in central and peripheral nervous system	Adrenocortical, breast, colorectal, leukemic, ovarian, and renal cancers
		ABCB5	GSH mediator	Leukemic, lung, melanoma, ovarian, renal, and thyroidal cancers
		ABCB8	Iron metabolism and homeostasis, OS protection	Head and neck, pancreatic, and renal cancers
	ABCC	ABCC1 (MRP1)	Organic anion transporter and GSH mediator	Breast, endometrium, glioma, head and neck, lung, lymphoma, melanoma, ovarian, prostate, neuroblastoma, and thyroid cancers
		ABCC2	Organic anion transporter	Colorectal, gastric, hepatic, and lung cancers
		ABCC3	Organic anion transporter	Breast, cervical, colorectal, gastric, hepatic, lung, ovarian pancreatic, renal, and thyroid cancers
		ABCC4	Nucleoside transporter	Breast, endometrial, gastric, head and neck, hepatic, lung, neuroblastoma, ovarian, prostate, andrenal cancers
		ABCC5	Nucleoside transporter	Breast, cervical, glioma hepatic, lung, pancreatic, renal, and urothelial cancers
		ABCC6	Putative biomineralization modulator	Liver cancer
		ABCC10	E(2)17βG transporter	Breast, colorectal, liver, lung, and prostate cancers
		ABCC11	Bile salts transporter	Breast cancer
		ABCC12	Unknown	Breast, colorectal, liver, lung, and prostate cancers
Type V	ABCG	ABCG2	Toxin efflux, cell differentiation	Cervical, glioma, liver, ovarian, prostate, pulmonary, renal, and testicular cancers
	ABCA	ABCA2	Lipid transporter	Breast, colon, leukemia, and liver cancers
		ABCA8	Lipophilic drugs transporter	Ovarian cancer
-	ABCF	ABCF2	Inflammatory development	Breast cancer

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
