# Peer review of "Drug Resistance and Endoplasmic Reticulum Stress in Hepatocellular Carcinoma"

_cells, 2022, doi:10.3390/cells11040632_

Round 1

Reviewer 1 Report

Overall, this review manuscript is a good paper, well conducted and organized and does add to our knowledge the state of the art about the involvement of ER-stress and activation of the UPR in the drug resistance during HCC progression. Nonetheless, some issues need to be still addressed.

The molecular mechanisms involved in drug resistance described in sections 3.1 to 3.5, should be graphically summarized in a picture showing the molecular interplay that contributes to chemoresistance in the HCC progression. According to the proposal of the authors, this seems to be the central contribution of the investigation but it was not graphically summarized. This will allow a better interpretation of summarized data.

The figure legend of figure 1 should explain in detail what in the image is shown. For example, PERK arm is not clearly pointed out in the image as was indicated for the other arms. All mechanisms shown in the image should be briefly explained.

This manuscript contains some typo issues that need to addressed; for examples, Elevated expression of transport proteins expressed at the plasma membrane LIMITS intracellular accumulation of anticancer drugs. So, the full manuscript should be carefully reviewed for grammar issues.

Author Response

Overall, this review manuscript is a good paper, well conducted and organized and does add to our knowledge the state of the art about the involvement of ER-stress and activation of the UPR in the drug resistance during HCC progression. Nonetheless, some issues need to be still addressed.

Thank you for your positive feedback. We have tried our best to address the suggestions provided below in the revised version of the manuscript. 

The molecular mechanisms involved in drug resistance described in sections 3.1 to 3.5, should be graphically summarized in a picture showing the molecular interplay that contributes to chemoresistance in the HCC progression. According to the proposal of the authors, this seems to be the central contribution of the investigation but it was not graphically summarized. This will allow a better interpretation of summarized data.

We have added a figure that summarizes the molecular mechanisms involved in drug resistance and also points out how the UPR is a central player in this process. This has now been included in the revised version of the manuscript. 

The figure legend of figure 1 should explain in detail what in the image is shown. For example, PERK arm is not clearly pointed out in the image as was indicated for the other arms. All mechanisms shown in the image should be briefly explained.

We have added PERK to the figure and updated the figure legend in the revised version of the manuscript.

This manuscript contains some typo issues that need to addressed; for examples, Elevated expression of transport proteins expressed at the plasma membrane LIMITS intracellular accumulation of anticancer drugs. So, the full manuscript should be carefully reviewed for grammar issues.

All authors have gone through the manuscript and adjusted language mistakes. 

Reviewer 2 Report

This review provides an overview of how ER-stress and activation of the UPR pathways with arms IRE1a, ATF6 and PERK contributes to drug resistance in HCC. The paper is well written and its focus is an important aspect of HCC cell biology.

As a point I would like to mention in the discussion of tumor microenvironment 2 recent papers on the subject ,discussed after line 447 of the manuscript

  1. Tumor-Associated Macrophages in Hepatocellular Carcinoma Pathogenesis, Prognosis and Therapy

Arvanitakis, K., Koletsa, T., Mitroulis, I., Germanidis, G. Cancers, 2022, 14(1), 226 2.

  1. Tumor-associated neutrophils in hepatocellular carcinoma pathogenesis, prognosis, and therapy

Arvanitakis, K., Mitroulis, I., Germanidis, G. Cancers, 2021, 13(12), 2899

The authors have to discuss if the 3 arms of UPR are influenced by the two types of cells of the tumor microenvironment

Author Response

This review provides an overview of how ER-stress and activation of the UPR pathways with arms IRE1a, ATF6 and PERK contributes to drug resistance in HCC. The paper is well written and its focus is an important aspect of HCC cell biology.

As a point I would like to mention in the discussion of tumor microenvironment 2 recent papers on the subject ,discussed after line 447 of the manuscript

  1. Tumor-Associated Macrophages in Hepatocellular Carcinoma Pathogenesis, Prognosis and Therapy, Arvanitakis, K., Koletsa, T., Mitroulis, I., Germanidis, G. Cancers, 2022, 14(1), 226 2.
  1. Tumor-associated neutrophils in hepatocellular carcinoma pathogenesis, prognosis, and therapy, Arvanitakis, K., Mitroulis, I., Germanidis, G. Cancers, 2021, 13(12), 2899

The authors have to discuss if the 3 arms of UPR are influenced by the two types of cells of the tumor microenvironment

Thank you for your positive feedback. These references have been added to the revised version of the manuscript and more information has been provided about the UPR and tumor-associated neutrophils.

Round 2

Reviewer 1 Report

The authors of the manuscript ID Cells-1549898-peer-review-v2, have rightly addressed my comments and suggestions. So, the corrected version of the manuscript is now acceptable for further steps in the publication process by Cells journal.